# OpenReview forum: "ShadowKV: KV Cache in Shadows for High-Throughput Long-Context LLM Inference"
_ICLR.cc/2025/Conference — Submitted to ICLR 2025_

### Official Review · Reviewer_iRxd · 2024-10-22

**Soundness:** 3
**Presentation:** 3
**Contribution:** 3
**Rating:** 8
**Confidence:** 3

**Summary:**

The paper introduces SHADOWKV, a novel system that improves inference throughput for long-context LLMs. The key challenge addressed is the increasing memory footprint of KV caches as sequence length increases, which slows down inference. SHADOWKV offers a solution by offloading the value cache to the CPU while keeping a low-rank key cache on the GPU, reducing memory consumption without sacrificing performance. By employing a method of KV selection for sparse attention, it boosts throughput and supports larger batch sizes, showing improvements of up to 3.04× throughput on the A100 GPU.

**Strengths:**

The paper tackles a significant problem in LLM inference by reducing GPU memory usage through a hybrid CPU-GPU cache system, enabling long-context LLMs to operate more efficiently without sacrificing accuracy.

The system shows impressive throughput gains, handling up to six times larger batch sizes compared to baseline methods, which could substantially impact real-world LLM deployment.

SHADOWKV is tested on multiple models and benchmarks (e.g., Llama, GLM) across various tasks, demonstrating consistent performance improvements across all settings.

**Weaknesses:**

The main issue is that, as far as I know, SVD is precision sensitive, but I didn't find any discussion about precision in the paper. My main question is what precision is used for ShadowKV and baselines. If you are using precision like FP16/FP32, my question is how does ShadowKV work on FP8/(FP8 & FP16) precision? If the precision is FP8, how does ShadowKV survive from precision-sensitive SVD?

For the speed evaluation, I only found the throughput (tokens/s), are there any experiments for the time of each operation separately?

**Questions:**

See weaknesses.

---

> ### Author Response · Authors · 2024-11-20
> **Response to Reviewer iRxd (Part 1/2)**
>
> Thank you for your encouraging feedback and insightful remarks. We have thoroughly addressed each of your questions and hope our responses will lead you to consider raising your score.
>
> **Q1: Precision Sensitivity in SVD Computations**
>
> We appreciate the suggestion to examine the precision sensitivity of ShadowKV. In the main experiments, we used BF16 for both model weights and KV cache. To further investigate the impact of precision on ShadowKV’s performance, we conducted additional experiments using FP8 precision (`torch.float8_e5m2`), showing that **ShadowKV can retain its accuracy at this lower precision**, addressing concerns about precision sensitivity, particularly in SVD computations (mentioned in $\textcolor{blue}{\text{Section 5.3, Page 10}}$ and detailed in  $\textcolor{blue}{\text{Section A.4, Page 16}}$).
>
> As detailed in the tables below, ShadowKV and baseline methods were evaluated using FP8. Results show that ShadowKV maintains accuracy and achieves consistently high performance even with FP8 precision. This robustness, despite FP8’s reduced numerical range, confirms that ShadowKV can continue to deliver efficiency gains without compromising accuracy.
>
> - Results on RULER
>
> |**Methods**|**N-S1**|**N-S2**|**N-MK1**|**N-MK2**|**N-MQ**|**N-MV**|**QA-1**|**QA-2**|**VT**|**FWE**|**Avg.**|
> |------------------------|-----------------|-----------------|----------------|----------------|----------------|----------------|----------------|----------------|----------------|----------------|----------------|
> |*Llama-3-8B-1M*|100.00|100.00|98.96|95.83|97.40|95.57|63.54|48.96|75.83|73.26|84.94|
> |Loki|5.21|1.04|0.00|0.00|0.78|0.26|5.21|13.54|28.33|28.82|8.32|
> |Loki (Vonly)|36.46|9.38|31.25|0.00|6.25|21.09|11.46|15.63|57.08|35.76|22.44|
> |Quest|**100.00**|98.96|**98.96**|71.88|96.61|**93.49**|63.54|45.83|78.13|67.01|81.44|
> |Quest (Vonly)|**100.00**|**100.00**|**98.96**|85.42|**97.40**|**93.49**|70.83|**48.96**|78.13|65.63|83.88|
> |**ShadowKV**|**100.00**|**100.00**|97.92|**94.79**|95.31|**93.49**|**75.00**|**48.96**|**80.42**|**73.61**|**85.95**|
>
> - Results on LongBench
>
> |**Methods**|**NarratQA**|**MultiFQA**|**HotpotQA**|**MuSiQue**|**DuRead**|**GovRep**|**SAMSum**|**PassRetr**|**LCC**|**Avg.**|
> |------------------------|----------------|----------------|----------------|----------------|----------------|----------------|----------------|----------------|----------------|----------------|
> |*Llama-3-8B-1M*|18.69|41.21|35.76|21.59|31.81|33.77|35.29|80.50|56.77|39.49|
> |Loki|2.21|11.12|5.70|1.84|15.42|28.59|11.41|41.91|33.99|16.91|
> |Loki (Vonly)|2.68|22.33|12.69|3.35|21.43|30.57|16.32|47.68|36.64|21.52|
> |Quest|19.41|38.92|34.02|19.64|23.13|26.40|28.04|78.50|49.81|35.32|
> |Quest (Vonly)|16.19|36.73|**36.64**|19.59|25.57|29.46|27.14|**79.50**|60.05|36.76|
> |**ShadowKV**|**18.29**|**39.39**|36.06|**21.04**|**30.47**|**31.87**|**35.56**|78.50|**62.11**|**39.25**|

---

> ### Author Response · Authors · 2024-11-20
> **Response to Reviewer iRxd (Part 2/2)**
>
> **Q2: Detailed Time Analysis of Operations**
>
> Thank you for the valuable suggestion. We present a detailed latency breakdown in tables below to illustrate the efficiency of each operation under various context lengths for both the prefilling and decoding stages (mentioned in $\textcolor{blue}{\text{Section 5.2, Page 9}}$ and detailed in  $\textcolor{blue}{\text{Appendix A.6, Page 18}}$).
>
> - **Scalability for Longer Sequences.** As shown in table, the overhead of SVD, reduce, cosine similarity, topK, and gather computing is very low and tends to decrease as the sequence scales, proving that ShadowKV's scalability to longer sequences.
>
> Latency breakdown (ms) of a Transformer block of Llama-3-8B-1M during prefilling:
>
> | **Context** | **Attention** | **FFN**    | **SVD**    | **Reduce** | **CosineSimilarity** | **TopK** | **Gather** | **Cost**   |
> |---------|-----------|--------|--------|--------|------------------|------|--------|--------|
> | 64K     | 186.23    | 96.47  | 17.19  | 0.10   | 1.41             | 0.08 | 0.01   | 6.65% |
> | 128K    | 721.13    | 193.32 | 26.62  | 0.20   | 2.77             | 0.14 | 0.02   | 3.25% |
> | 256K    | 2880.21   | 392.77 | 50.56  | 0.42   | 6.11             | 0.11 | 0.03   | 1.75% |
> | 512K    | 11720.30  | 789.23 | 108.38 | 0.84   | 12.19            | 0.15 | 0.06   | 0.97% |
>
> - **Overlapping Operations for Latency Reduction.** In the table below, we demonstrate how overlapping the recomputation of the key cache with value cache fetching from the CPU significantly reduces decoding latency. This concurrent processing approach ensures that ShadowKV minimizes overhead when handling long-context models.
>
> Latency breakdown (ms) of a Transformer block of Llama-3-8B-1M during decoding:
>
> |**Context**|**GEMM+Softmax**|**Max**|**TopK**|**Recompute K (Overlapped)**|**Fetch V**|**Attention**|**FFN**|**QKV**|
> |----------------|---------------------------------|-----------|------------|--------------------------------------|---------------|-----------------|-----------|-----------|
> |48$\times$64K|0.56|0.07|0.14|1.25|1.84|0.23|0.33|0.05|
> |24$\times$128K|0.58|0.07|0.15|1.36|1.66|0.21|0.29|0.05|
> |12$\times$256K|0.65|0.07|0.16|1.49|1.75|0.19|0.25|0.05|
> |6$\times$512K|0.71|0.07|0.17|1.51|1.69|0.18|0.23|0.05|

---

> > ### Author Response · Authors · 2024-11-25
> > **Grateful for Your Insights and Follow-Up on Our Responses for Reviewer iRxd**
> >
> > Dear Reviewer iRxd,
> >
> > We sincerely appreciate the time and effort you have dedicated to reviewing our work and for sharing your insightful feedback. Your comments have been immensely valuable in improving the quality of our manuscript.
> >
> > We have carefully addressed your concerns and revised the manuscript accordingly. As the discussion period is ending soon, we wanted to kindly ask if there are any further questions or points we could clarify to better address your feedback. If our responses have adequately addressed your concerns, we would be deeply grateful if you could consider reflecting this in your score.
> >
> > Thank you again for your constructive review and for contributing your valuable expertise. We deeply appreciate your input and support throughout this process.

---

> ### Author Response · Authors · 2024-11-28
> **Follow Up Before PDF Revision Deadline**
>
> Dear Reviewer iRxd,
>
> We are truly grateful for the time and effort you have taken to review our manuscript and for providing such thoughtful feedback. Your insights have been instrumental in helping us refine our work.
>
> As today marks the final day for revising the PDF, we wanted to kindly follow up to see if you have any further questions or points of clarification regarding our responses. We are happy to address any remaining concerns you might have.
>
> Thank you again for your invaluable support and guidance throughout this process.
>
> With our deepest appreciation,
>
> Authors

---

> > ### Comment · Reviewer_iRxd · 2024-11-30
> >
> > What is the precision for SVD? PyTorch's SVD seems to not support FP16 or lower precision. See [link](https://github.com/NVIDIA/apex/issues/887).

---

> ### Author Response · Authors · 2024-11-30
>
> Thank you for your thoughtful question and for pointing this out. The SVD operation is conducted in FP32, as PyTorch currently does not support lower-precision kernels for SVD. To evaluate the robustness of ShadowKV, **we cast the decomposition results to FP8, optimizing KV cache storage while ensuring performance remains robust**. During decoding, the K reconstruction (i.e., matrix multiplication) is performed directly in FP8 or BF16, just like other FP8 or BF16 LLM decoding operations. Moreover, this SVD part can be **executed on the CPU during prefilling, which can be overlapped**, mitigating the overhead.
>
> Additionally, we explored PyTorch's approximation SVD method, `gesvda`, and found that it does not degrade ShadowKV's performance, further demonstrating its precision robustness. Once PyTorch introduces lower-precision SVD kernels, we would be happy to evaluate their impact on our method.
>
> If our responses have adequately addressed your concerns, we would be deeply grateful if you could consider reflecting this in your score. Please let me know if you have any further questions or suggestions.

---

> > ### Author Response · Authors · 2024-12-02
> > **Follow Up Before Deadline**
> >
> > Dear Reviewer iRxd,
> >
> > We sincerely appreciate the time and effort you have dedicated to reviewing our manuscript and for sharing your thoughtful feedback. Your insights have been invaluable in helping us improve our work.
> >
> > As today marks the final day for reviewer replies, we wanted to kindly follow up to check if you have any additional questions or require further clarification regarding our responses. We would be more than happy to address any remaining concerns you may have.
> >
> > Thank you once again for your invaluable support and guidance throughout this process.
> >
> > With our deepest gratitude,
> >
> > The Authors

---

### Official Review · Reviewer_5wSX · 2024-10-31

**Soundness:** 2
**Presentation:** 1
**Contribution:** 3
**Rating:** 5
**Confidence:** 4

**Summary:**

This paper introduces SHADOWKV, a CPU-offloading-based system for long-context LLM inference. SHADOWKV addresses GPU memory constraints by leveraging the low-rank property of the key cache, storing a low-rank pre-ROPE key cache on the GPU while offloading the value cache to the CPU. Additionally, SHADOWKV stores landmarks—mean values of post-ROPE keys for adjacent tokens—along with low-rank pre-ROPE key cache in GPU memory. During decoding, SHADOWKV utilizes these landmark keys to identify significant tokens, selectively recovering their keys and retrieving their values from the CPU. Evaluations on long-context benchmarks demonstrate that SHADOWKV maintains high accuracy while significantly reducing GPU memory consumption and inference latency.

**Strengths:**

* In-depth analysis on low-rank nature of key cache in comparison to value cache as well as weights.
* Leveraging spatial locality of post-ROPE key cache for dynamic sparse attention appears both novel and effective.
* Empirical results are impressive.

**Weaknesses:**

* Incomplete Descriptions

The term "sparse budget" is not clearly defined in the paper, which may lead to confusion. Additionally, while SHADOWKV claims to leverage the temporal locality of the KV cache to reduce computation and communication (by approximately 60%), it lacks any detailed explanations on what that feature is.

* Handling Newly Generated Tokens

While the paper says that it excludes the handling of newly generated tokens for simplicity, this issue is quite significant and should not be ignored. If not addressed, the KV cache for newly generated tokens could negate SHADOWKV’s key benefits of reduced GPU memory usage and lower inference latency, especially with long output sequences. Incorporating mechanisms to handle these tokens within SHADOWKV is essential, and the authors should evaluate and report on its impact on accuracy.

* Lack of Comparison with Infinigen

The paper does not sufficiently compare SHADOWKV with Infinigen [1], a closely related work that similarly stores low-rank key cache in GPU memory, offloads the value cache to the CPU, and selectively fetches important values based on approximate attention scores. Although the paper briefly discusses Infinigen, given the significant similarities, a more in-depth comparison with Infinigen should be made in order to highlight the main differentiator of SHADOWKV.

[1] Lee et al., "InfiniGen: Efficient Generative Inference of Large Language Models with Dynamic KV Cache Management", OSDI'24

**Questions:**

* What exactly is the “sparse budget”?

* How does SHADOWKV leverage the temporal locality of the KV cache?

* What are the exact GPU memory savings of SHADOWKV? Including a quantitative discussion on GPU memory savings in the paper would be helpful.

* How can SHADOWKV handle the KV cache for newly generated tokens, and what would be the impact of this?

---

> ### Author Response · Authors · 2024-11-20
> **Response to Reviewer 5wSX (Part 1/4)**
>
> Thank you sincerely for your thoughtful review and invaluable feedback. We have addressed each of your questions with care and hope that our responses will inspire you to consider raising your score.
>
> **Q1: Definition of “sparse budget”**
>
> Thanks for pointing it out. The term “sparse budget” refers to the number of selected tokens, i.e. the K of TopK. This budget dictates the portions that need to be fetched from the CPU or reconstructed from the low-rank space.  We have added this clarification to the revised paper ($\textcolor{blue}{\text{Section 1, Page 2}}$).
>
> ---
>
> **Q2:  Explanation of “temporal locality of the KV cache”**
>
> Temporal locality of the KV cache refers to the observation that, during decoding, the KV cache pairs selected by the queries of two adjacent decoding steps have a repetition rate of approximately 60%. This means that we don’t need to perform low-rank reconstruction or CPU data fetching for the repeated portions, only for the non-repeated parts. This reduces the overall decoding overhead.  We have added this clarification to the revised paper ($\textcolor{blue}{\text{Section 3.2, Page 5}}$).

---

> ### Author Response · Authors · 2024-11-20
> **Response to Reviewer 5wSX (Part 2/4)**
>
> **Q3: Handling of Newly Generated Tokens**
>
> We appreciate the suggestion to evaluate the impact of handling newly generated tokens. We present extensive experiments on the RULER and LongBench across different long-context models (mentioned in $\textcolor{blue}{\text{Section 4.1, Page 6}}$ and detailed in  $\textcolor{blue}{\text{Appendix A.1, Page 15}}$).  **The results demonstrate that ShadowKV effectively handles newly generated tokens while maintaining accuracy.**
>
> To address the handling of newly generated tokens, we project these tokens' key cache into a low-rank space using the same projections applied during the prefilling phase. This approach preserves the benefits of reduced GPU memory usage, particularly for long output sequences.
>
> As shown in tables below, we refer to this extension as ShadowKV+. Our evaluation across various models demonstrates that ShadowKV+ effectively maintains accuracy and manages newly generated tokens as well.
>
> - Results on RULER:
>
> | **Methods** | **N-S1** | **N-S2** | **N-MK1** | **N-MK2** | N-MQ | **N-MV** | **QA-1** | **QA-2** | **VT** | **FWE** | **Avg.** |
> |---|---|---|---|---|---|---|---|---|---|---|---|
> | *Llama-3-8B-1M* | 100.00 | 100.00 | 98.96 | 98.96 | 98.96 | 95.57 | 75.00 | 48.96 | 78.54 | 71.85 | 86.68 |
> | ShadowKV | **100.00** | **100.00** | 97.92 | 98.96 | 96.88 | **95.83** | **72.92** | **52.08** | **81.67** | **72.57** | **86.88** |
> | **ShadowKV+** | **100.00** | **100.00** | **98.96** | **100.00** | 95.83 | 93.49 | 71.88 | 50.00 | 80.21 | 71.88 | 86.23 |
> | *GLM-4-9B-1M* | 100.00 | 100.00 | 94.79 | 87.50 | 99.74 | 93.75 | 67.71 | 55.21 | 97.29 | 72.22 | 86.82 |
> | ShadowKV | **100.00** | **100.00** | **95.83** | 83.33 | **98.70** | **87.76** | **69.79** | 55.21 | 97.50 | **68.06** | 85.62 |
> | **ShadowKV+** | **100.00** | **100.00** | **95.83** | **85.42** | 98.17 | 85.16 | **69.79** | **56.25** | **97.92** | 67.71 | **85.63** |
> | *Llama-3.1-8B* | 100.00 | 100.00 | 98.96 | 91.67 | 98.96 | 95.31 | 82.29 | 47.92 | 68.96 | 71.18 | 85.53 |
> | ShadowKV | **100.00** | **100.00** | **100.00** | 83.33 | **97.92** | **92.19** | **81.25** | 48.96 | **67.08** | **64.93** | **83.57** |
> | **ShadowKV+** | **100.00** | **100.00** | **100.00** | **84.38** | 96.88 | 91.67 | **81.25** | **52.08** | 65.63 | 62.85 | 83.47 |
> | *Yi-9B-200K* | 100.00 | 100.00 | 86.46 | 62.50 | 64.58 | 32.55 | 44.79 | 39.58 | 36.87 | 89.93 | 65.73 |
> | ShadowKV | **100.00** | **100.00** | **82.29** | **67.71** | **63.28** | 31.51 | 43.75 | **38.54** | **56.04** | 72.22 | **65.53** |
> | **ShadowKV+** | **100.00** | **100.00** | 81.25 | **67.71** | 61.72 | 31.51 | **46.88** | **38.54** | 53.54 | **72.92** | 65.41 |
>
>
> - Results on LongBench:
>
> |**Methods**|**NarratQA**|**MultiFQA**|**HotpotQA**|**MuSiQue**|DuRead|**GovRep**|**SAMSum**|**PassRetr**|**LCC**|**Avg.**|
> |---|---|---|---|---|---|---|---|---|---|---|
> |*Llama-3-8B-1M*|18.98|41.84|36.79|21.47|31.93|34.18|35.96|81.50|56.07|39.86|
> |ShadowKV|17.17|39.73|**38.29**|**21.08**|**31.77**|31.62|**35.87**|**80.00**|**63.93**|39.94|
> |**ShadowKV+**|**20.42**|**41.16**|37.22|21.03|**31.77**|**31.98**|35.80|**80.00**|63.89|**40.36**|
> |*GLM-4-9B-1M*|25.44|51.09|58.67|39.61|32.04|29.97|40.31|99.00|58.02|48.24|
> |ShadowKV|26.50|**51.31**|59.09|**38.87**|32.92|28.54|38.70|**96.50**|**58.55**|47.89|
> |ShadowKV+|**27.59**|**51.31**|**59.17**|38.34|**33.55**|**31.25**|**39.46**|**96.50**|55.86|**48.11**|
> |*Llama-3.1-8B*|31.56|55.10|57.65|29.46|35.26|34.45|29.84|100.00|67.31|48.96|
> |ShadowKV|30.93|**55.20**|57.32|**29.13**|**31.85**|32.79|**30.40**|**99.50**|66.03|**48.13**|
> |**ShadowKV+**|**32.25**|54.29|**57.75**|28.37|31.07|**32.89**|28.73|98.75|**67.59**|47.97|
> |*Yi-9B-200K*|13.88|30.02|52.46|28.20|22.29|30.25|19.08|67.00|73.50|37.41|
> |ShadowKV|12.44|30.82|**52.43**|**27.73**|**20.79**|**29.83**|20.73|64.00|72.89|36.85|
> |**ShadowKV+**|14.08|**30.94**|51.16|27.00|19.50|29.34|**21.16**|**66.00**|**73.47**|**36.96**|

---

> ### Author Response · Authors · 2024-11-20
> **Response to Reviewer 5wSX (Part 3/4)**
>
> **Q4: Detailed Comparison with InfiniGen**
>
> We provide further clarification on the key distinctions and conduct additional experiments between ShadowKV and InfiniGen (mentioned in $\textcolor{blue}{\text{Section 2, Page 3}}$ and detailed in  $\textcolor{blue}{\text{Section 5.1, Page 7-8}}$).  **These experiments show that ShadowKV significantly outperforms InfiniGen across a wide range of downstream tasks.**
>
> - **Differences in SVD Usage.** Infinigen performs an offline SVD to get a projection matrix, which is applied to post-RoPE key and query states for KV selection, while ShadowKV applies an online, prompt-dependent SVD directly to the pre-RoPE key cache for compression, not for KV selection.
>
> - **Methodological Differences.** While InfiniGen uses SVD for KV selection, it requires fetching selected, exact KV pairs from the CPU. In contrast, ShadowKV only fetches the value cache from the CPU, reconstructing the key cache from its low-rank storage on the GPU. By overlapping these processes, ShadowKV reduces data-fetch overhead and achieves improved efficiency in KV cache management.
>
> - **Accuracy Comparison.** To empirically validate ShadowKV’s advantages, we conducted accuracy evaluations. Results, presented in the tables below, confirm ShadowKV’s effectiveness in maintaining accuracy while optimizing memory usage. Although InfiniGen performs well on simpler tasks like RULER-N-S1, it shows significant accuracy drops on more complex tasks, such as RULER-N-MK2, RULER-FWE, LongBench-LCC, and others, where ShadowKV maintains consistently high accuracy.
>
> - Results on RULER:
>
> |**Methods**|**N-S1**|**N-S2**|**N-MK1**|**N-MK2**|**N-MQ**|**N-MV**|**QA-1**|**QA-2**|**VT**|**FWE**|**Avg.**|
> |---|---|---|---|---|---|---|---|---|---|---|---|
> |*Llama-3-8B-1M*|100.00|100.00|98.96|98.96|98.96|95.57|75.00|48.96|78.54|71.85|86.68|
> |InfiniGen|**100.00**|98.96|84.38|53.13|63.28|54.95|65.63|48.96|**81.67**|50.35|70.13|
> |InfiniGen (Vonly)|**100.00**|98.96|96.88|76.04|81.25|77.08|67.71|50.00|**81.67**|53.47|78.31|
> |**ShadowKV**|**100.00**|**100.00**|**97.92**|**98.96**|**96.88**|**95.83**|**72.92**|**52.08**|**81.67**|**72.57**|**86.88**|
> |*GLM-4-9B-1M*|100.00|100.00|94.79|87.50|99.74|93.75|67.71|55.21|97.29|72.22|86.82|
> |InfiniGen|**100.00**|93.75|82.29|0.00|79.43|60.16|57.29|53.13|92.71|57.29|67.60|
> |InfiniGen (Vonly)|**100.00**|96.88|87.50|7.29|95.31|75.26|56.25|54.17|95.63|60.76|72.91|
> |**ShadowKV**|**100.00**|**100.00**|**95.83**|**83.33**|**98.70**|**87.76**|**69.79**|**55.21**|**97.50**|**68.06**|**85.62**|
> |*Llama-3.1-8B*|100.00|100.00|98.96|91.67|98.96|95.31|82.29|47.92|68.96|71.18|85.53|
> |InfiniGen|**100.00**|77.08|78.13|13.54|58.07|47.40|65.63|41.67|60.83|50.35|59.27|
> |InfiniGen (Vonly)|**100.00**|88.54|87.50|26.04|79.43|77.08|72.92|43.75|57.08|55.21|68.76|
> |**ShadowKV**|**100.00**|**100.00**|**100.00**|**83.33**|**97.92**|**92.19**|**81.25**|**48.96**|**67.08**|**64.93**|**83.57**|
> |*Yi-9B-200K*|100.00|100.00|86.46|62.50|64.58|32.55|44.79|39.58|36.87|89.93|65.73|
> |InfiniGen|**100.00**|94.79|77.08|1.04|40.10|20.57|37.50|34.38|41.46|46.18|49.31|
> |InfiniGen (Vonly)|**100.00**|98.96|78.13|2.08|58.33|24.48|40.63|35.42|52.92|55.90|54.69|
> |**ShadowKV**|**100.00**|**100.00**|**82.29**|**67.71**|**63.28**|31.51|**43.75**|**38.54**|**56.04**|**72.22**|**65.53**|
>
> - Results on LongBench:
>
> |**Methods**|**NarratQA**|**MultiFQA**|**HotpotQA**|**MuSiQue**|**DuRead**|**GovRep**|**SAMSum**|**PassRetr**|**LCC**|**Avg.**|
> |---|---|---|---|---|---|---|---|---|---|---|
> |*Llama-3-8B-1M*|18.98|41.84|36.79|21.47|31.93|34.18|35.96|81.50|56.07|39.86|
> |InfiniGen|14.39|31.46|33.63|17.94|26.65|27.38|21.97|74.30|38.58|31.81|
> |InfiniGen (Vonly)|*17.83*|36.08|35.28|19.64|28.39|29.28|28.12|74.85|45.53|35.00|
> |**ShadowKV**|17.17|*39.73*|*38.29*|*21.08*|**31.77**|**31.62**|**35.87**|**80.00**|**63.93**|**39.94**|
> |*GLM-4-9B-1M*|25.44|51.09|58.67|39.61|32.04|29.97|40.31|99.00|58.02|48.24|
> |InfiniGen|23.67|46.31|55.69|33.91|27.49|25.44|33.48|91.83|36.96|41.64|
> |InfiniGen (Vonly)|25.63|48.44|57.23|36.94|29.77|26.67|36.64|93.58|46.69|44.62|
> |**ShadowKV**|**26.50**|**51.31**|**59.09**|**38.87**|**32.92**|**28.54**|**38.70**|**96.50**|**58.55**|**47.89**|
> |*Llama-3.1-8B*|31.56|55.10|57.65|29.46|35.26|34.45|29.84|100.00|67.31|48.96|
> |InfiniGen|27.23|52.72|53.89|26.81|27.72|29.61|24.42|98.93|56.89|44.25|
> |InfiniGen (Vonly)|29.73|53.47|55.11|28.72|28.55|31.42|26.76|99.17|62.66|46.18|
> |**ShadowKV**|**30.93**|**55.20**|**57.32**|**29.13**|**31.85**|**32.79**|**30.40**|**99.50**|**66.03**|**48.13**|
> |*Yi-9B-200K*|13.88|30.02|52.46|28.20|22.29|30.25|19.08|67.00|73.50|37.41|
> |InfiniGen|10.01|23.61|50.47|25.91|15.11|27.96|18.97|30.00|56.46|28.72|
> |InfiniGen (Vonly)|11.31|26.46|51.13|26.77|16.09|28.67|19.33|34.00|62.07|30.65|
> |**ShadowKV**|**12.44**|**30.82**|**52.43**|**27.73**|**20.79**|**29.83**|**20.73**|**64.00**|**72.89**|**36.85**|

---

> > ### Author Response · Authors · 2024-11-20
> > **Response to Reviewer 5wSX (Part 4/4)**
> >
> > **Q5:  Quantitative Analysis of GPU Memory Savings**
> >
> > Thank you for the valuable suggestion. The GPU memory savings provided by ShadowKV can be quantitatively analyzed as follows (mentioned in $\textcolor{blue}{\text{Section 4.2, Page 6}}$ and detailed in  $\textcolor{blue}{\text{Section A.2, Page 16}}$). Let each K or V vector have a size of $M$ bytes, with a sequence length $S$, a chunk size $C$, a selected chunk budget $K$, $O$ outliers, and a pre-RoPE key cache rank $r$. The GPU memory savings of ShadowKV can then be expressed as:
> >
> > $$
> >     \text{Memory Savings} = \frac{2SM}{SM/C + 2(K+O)C+ Sr+rM}
> > $$
> >
> > For example, assuming $M=1024, C=8, S=128\text{K}, K=256, O=48, r=160$, the memory savings of ShadowKV is calculated as **7.08$\times$**. This result demonstrates that ShadowKV can theoretically reduce the KV cache memory footprint on the GPU by **7.08$\times$** for longer sequences and larger batch sizes.

---

> > > ### Comment · Reviewer_5wSX · 2024-11-22
> > >
> > > Thank you for the responses. Before taking a closer look into the additional experimental data you’ve provided, I have a quick question regarding the GPU memory savings. Does your quantitative analysis of GPU memory savings account for the way temporal locality is leveraged? It might increase GPU memory consumption.

---

> > ### Comment · Reviewer_5wSX · 2024-11-23
> >
> > Thank you for the additional experimental data. I really appreciate that you were able to provide all this information within such a short time frame.
> >
> > By the way, while you compare ShadowKV and InfiniGen in terms of accuracy, I believe there should be a discussion about latency and GPU memory savings for a proper comparison between the two methods. Specifically, I am curious whether the latency and GPU memory savings for these methods were set to the same level.
> >
> > It would be great if you could provide some details on this aspect.

---

> ### Author Response · Authors · 2024-11-22
> **Clarification about quantitative analysis of GPU memory savings**
>
> Hello, thank you for your response. Our cache implementation is actually quite straightforward—we avoid complex mechanisms like LRU, ensuring no additional memory consumption. Specifically, during each decoding step, we overwrite the KV pairs from the previous step **using the same buffer** ($2(K+O)C$ in the formula), effectively reusing memory without allocating extra space.
>
> If more complex cache rules are used, the hit rate can be improved, but it will bring some additional memory overhead.

---

> > ### Comment · Reviewer_5wSX · 2024-11-22
> >
> > Thank you for the prompt response. I understand your point, but I have one more quick question. I am a bit confused about the notation. What exactly is the selected chunk budget (K)? Is it equivalent to the KV cache budget or the sparse budget mentioned in your evaluation?
> >
> > For instance, in the description of the settings for the LongBench evaluation in the paper, it is stated that the KV cache budget is set to 256. I wonder if the KV cache budget in this context corresponds to the selected chunk budget in the GPU memory savings analysis.

---

> ### Author Response · Authors · 2024-11-22
> **Clarification about KV budget**
>
> Thanks for your response. The **sparse budget** is calculated as the product of the **selected chunk budget (K)** and the **chunk size (C)**, i.e., $\text{sparse budget} = K \times C$.
>
> For LongBench, when we say “KV cache budget is set to 256,” it means the sparse budget is 256. If the chunk size is 8, then $K = 256/8=32$.
>
> Feel free to reach out if you have any additional questions. Thanks!

---

> > ### Comment · Reviewer_5wSX · 2024-11-22
> >
> > Thank you for the clarification. Just to confirm, the 7.08x memory savings apply specifically to the example setting used in your quantitative analysis, and GPU memory usage varies across benchmarks in accuracy evaluation due to the different sparse budgets set for each benchmark, correct? The way you set the sparse budget differently for each benchmark makes it a bit confusing to exactly grasp memory saving-accuracy tradeoff of your scheme.
> >
> > That said, I now have a much clearer understanding of this point. Thank you again for the explanation.

---

> ### Author Response · Authors · 2024-11-22
>
> Thank you for your feedback! I'm very glad the explanation clarified things for you. Yes, you are correct. In the efficiency test, we demonstrate that ShadowKV achieves 6x larger batch sizes (refer to $\textcolor{blue}{\text{Table 4}}$), which aligns well with our theoretical analysis.
>
> For RULER-128K, we use a budget of  $K = 256$ . For LongBench, since the context lengths are shorter ($\le$32K), we opt for a smaller $K$ to better match the benchmark’s requirements.
>
> Feel free to reach out if you have any additional questions. Thanks again for your engagement!

---

> ### Author Response · Authors · 2024-11-23
>
> Thank you for following up and for your thoughtful comments. We appreciate the opportunity to provide additional details regarding the comparison between ShadowKV and InfiniGen in terms of GPU memory savings and latency. Below, we address your questions and provide a detailed analysis, demonstrating that **ShadowKV outperforms InfiniGen in terms of accuracy, memory savings, and latency**.
>
> **GPU memory savings:** InfiniGen does not provide a quantitative analysis of GPU memory savings in its paper, so we performed an estimation to highlight ShadowKV’s advantages. Following the configuration described in the InfiniGen paper, where the partial weight ratio is set to 0.3, the KV cache size is reduced to 15%, **equating to a $6.67\times$ memory savings. In comparison, ShadowKV achieves a $7.08\times$ reduction**. It should be noted that InfiniGen’s 15% memory savings does not account for additional memory overheads (e.g., KV buffers). If these overheads are included, the memory savings for InfiniGen would be further diminished.
>
> **Latency Analysis:** **From a latency perspective, ShadowKV is more scalable to long sequences than InfiniGen**. In our accuracy evaluation, we maintained the same sparse budget for both methods. Below is the latency breakdown (in ms) for a single Transformer block of Llama-3-8B-1M during decoding, using ShadowKV.
>
> | **Context**    | **GEMM+Softmax** | **Max** | **TopK** | **Recompute K (Overlapped)** | **Fetch V** | **Attention** | **FFN** | **QKV** |
> | -------------- | ---------------- | ------- | -------- | ---------------------------- | ----------- | ------------- | ------- | ------- |
> | 48$\times$64K  | 0.56             | 0.07    | 0.14     | 1.25                         | 1.84        | 0.23          | 0.33    | 0.05    |
> | 24$\times$128K | 0.58             | 0.07    | 0.15     | 1.36                         | 1.66        | 0.21          | 0.29    | 0.05    |
> | 12$\times$256K | 0.65             | 0.07    | 0.16     | 1.49                         | 1.75        | 0.19          | 0.25    | 0.05    |
> | 6$\times$512K  | 0.71             | 0.07    | 0.17     | 1.51                         | 1.69        | 0.18          | 0.23    | 0.05    |
>
> ShadowKV overlaps the recomputation of the K cache with the retrieval of the V cache from CPU. In contrast, InfiniGen overlaps the KV cache fetching time with other computations (GEMM, Softmax, Max, TopK, Attention, FFN, and QKV operations).
> **However, as the sequence length increases, InfiniGen's ability to hide data-fetching costs diminishes.** For instance, at a sequence length of $12 \times 256K$, the cost of other computations is 1.37 ms (0.65+0.07+0.16+0.19+0.25+0.05):
>
>  - For ShadowKV, the latency is $1.75+1.37=3.12\text{ms}$, as K cache recomputation is effectively overlapped.
>  - For InfiniGen, the KV cache fetching time alone is already $1.75\times 2=3.5\text{ms}$ (InfiniGen fetches both K and V, while ShadowKV only fetches V from CPU). Thus, at longer sequence lengths, the other computations in InfiniGen cannot sufficiently hide the data-fetching cost.
>
> Therefore, under the same sparse budget, **ShadowKV not only outperforms InfiniGen in latency but also achieves significantly better accuracy on complex tasks, where InfiniGen suffers from a notable accuracy drop.**
>
> We hope this analysis addresses your concerns and provides a comprehensive understanding of ShadowKV's effectiveness. Thank you for your insightful feedback, and we remain open to any further questions or discussions.

---

> > ### Comment · Reviewer_5wSX · 2024-11-23
> >
> > Thank you for your prompt response.
> >
> > I have some reservations about the claim that ShadowKV achieves a 7.08x memory savings when you compare it with InfiniGen. As I mentioned earlier, this figure seems specific to the particular setting used for its derivation (K=256, sequence length=128K). Meanwhile, for LongBench, you use K=32, which implies that a sequence length should be somewhere around 16K to achieve that level of GPU memory savings. However, as far as I know, the sequence lengths of LongBench are generally much shorter than that, which could result in lower GPU memory savings.
> >
> > For this perspective, I feel the comparison with InfiniGen may not be fully fair. If I’ve misunderstood something, I’d appreciate any clarification.
> >
> > By the way, I’m also having trouble understanding the statement: “InfiniGen’s 15% memory savings does not account for additional memory overheads (e.g., KV buffers).” Could you elaborate on what “KV buffers” refers to in this context?

---

> ### Author Response · Authors · 2024-11-23
>
> Thank you for your thoughtful feedback and for pointing out your concerns. Let me clarify the points you raised.
>
> The term "KV buffers" refers to the $2KC$, i.e., sparse budget component in this context (for LongBench, we set $O = 1$ for ShadowKV). To implement a caching mechanism, **both InfiniGen and ShadowKV must retain at least this portion of the memory, which reduces the overall memory savings**. Therefore, your concern about the effect of $K$ on memory savings applies equally to InfiniGen. Moreover, InfiniGen introduces **additional memory overhead by saving the low-rank projection matrix**, which is also not accounted for in the $6.67\times$ memory savings.
>
> **Our primary goal with ShadowKV is to optimize for long-context scenarios ($S >16K$, benchmarks like RULER), where its scalability becomes more evident.** To provide additional clarity, we have included a table showing memory savings for ShadowKV across different sequence lengths $S$ and $K = 32$. It’s worth noting that, for the LongBench dataset, we only evaluate samples with sequence lengths greater than $4K$.
>
> As shown in the table below, ShadowKV matches InfiniGen's memory savings (which is overestimated) at a sequence length of $8K$ and surpasses it as the sequence length increases. We believe that **ShadowKV demonstrates better performance in long-context scenarios compared to InfiniGen and offers superior scalability.**
>
> | **(S, K)**       | **S = 4K, K = 32** | **S = 8K, K = 32** | **S = 16K, K = 32** | **S = 32K, K = 32** |
> |-------------------|--------------------|--------------------|---------------------|---------------------|
> | **Memory Savings** | 6.24$\times$             | 6.65$\times$              | 6.87$\times$               | 6.99$\times$               |
>
> We hope this table and explanation address your concerns and provide further insight into ShadowKV's scalability and performance. Please feel free to reach out with any additional questions or clarifications. We are looking forward to your feedback.

---

> > ### Comment · Reviewer_5wSX · 2024-11-25
> >
> > I'm now clear with this point. Thank you for the detailed explanations.

---

> > > ### Author Response · Authors · 2024-11-25
> > >
> > > Thank you so much for your kind and detailed feedback. We are deeply grateful for the time and effort you’ve devoted to reviewing our work and for providing us with such valuable comments.
> > >
> > > It means a lot to us that our explanations addressed your concerns, and we sincerely hope that you might consider raising your score to reflect your clarified understanding.
> > >
> > > Your support and constructive feedback are truly appreciated, and please don’t hesitate to let us know if there’s anything more we can do. Thank you again for your generosity and guidance.

---

> > > > ### Author Response · Authors · 2024-11-28
> > > > **Final Follow Up Before PDF Revision Deadline**
> > > >
> > > > Dear Reviewer 5wSX,
> > > >
> > > > We are truly grateful for the time and effort you have taken to review our manuscript and for providing such thoughtful feedback. Your insights have been instrumental in helping us refine our work.
> > > >
> > > > As today marks the final day for revising the PDF, we wanted to kindly follow up to see if you have any further questions or points of clarification regarding our responses. We are happy to address any remaining concerns you might have.
> > > >
> > > > Thank you again for your invaluable support and guidance throughout this process.
> > > >
> > > > With our deepest appreciation,
> > > >
> > > > Authors

---

### Official Review · Reviewer_4R5o · 2024-11-02

**Soundness:** 2
**Presentation:** 3
**Contribution:** 3
**Rating:** 6
**Confidence:** 3

**Summary:**

The increasing KV-cache poses great challenges to long-context LLM inference. This work presents a long-context LLM inference system, SHADOWKV. It decreases the memory footprint by storing the low-rank key cache and offloading value cache to CPU, and reduces the decoding latency by reconstructing the sparse KV pairs on-the-fly. Evaluations show that SHADOWKV supports up to 6x larger batch sizes and improves throughput by up to 3X on A100 GPU while maintaining the accuracy.

**Strengths:**

+ This work presents a very interesting observation on the KV cache: pre-RoPE key cache is exceptionally low-rank compared to post-RoPE key cache, value cache and KV projection weights. Built on this observation, the proposed SHADOWKV significantly reduces the memory footprint of the Key cache.
+ The work also improves the previous sparse attention work including QUEST by introducing outlier KV cache.
+ This work also implements the inference system and shows actual throughput improvement on the real world A100 GPUs.

**Weaknesses:**

- The proposed method is complex, including low-rank K cache, CPU offloaded V cache, outlier KV cache, and dynamic sparse attention. However,  the ablation study on each component is missing.
   - In terms of model accuracy:
      - it is unclear how much accuracy improvement an extra outlier KV cache will bring.
      - previous work Quest uses Min-Max as landmark cache, ShadowKV adopts Mean as landmark cache. It is unclear how much accuracy improvement this change will bring.
   - In terms of efficiency:
      - Authors only show a rough prefiling latency breakdown in Figure 1(c). It is unclear unclear how long it takes for computing the outlier cache (i.e., reduce, cosine-similarity, top-k, gather) in the profiling stage, how long it takes for KV cache chunk selection (i.e, MatMul, Softmax, Max, TopK) in the decoding stage, how long it takes for recomputing the K cache from low-rank cache. These overheads seem to increase linearly with the context length. It would be better to see the efficiency breakdown on every part of the system under different context lengths (e.g., 128K, 256K, 512K).
      - it is unclear how SHADOWKV performs for extremely long context. For example, the authors evaluated on Llama-3-8B-1M but only with up to 128K context length.
     - this work lacks efficiency comparison against previous work LoKi, Quest and MInference.

**Questions:**

My questions are listed in the weakness section.

---

> ### Author Response · Authors · 2024-11-20
> **Response to Reviewer 4R5o (Part 1/4)**
>
> Thank you for detailed review and valuable feedback. We appreciate the reviewer’s recognition of the novelty and effectiveness of our method. Below, we address concerns regarding the lack of some ablation results. We hope the reviewer can consider raising your score in light of our response.
>
> **Q1: Accuracy Contribution of Outlier KV Cache and Mean Landmarks**
>
> Our additional experiments demonstrate that outliers play a critical role in capturing essential information, even in small numbers (0.049%), significantly enhancing the performance of mean-based landmarks and outperforming the min-max landmarks used in Quest.
>
> We conduct experiments using different numbers of outlier chunks for Llama-3-8B-1M on the RULER benchmark with 128K context length (mentioned in $\textcolor{blue}{\text{Section 5.3, Page 10}}$ and detailed in  $\textcolor{blue}{\text{Appendix A.8, Page 19}}$). As presented in the table below, our findings indicate that outliers play a crucial role. For instance, the first chunk, a significant outlier, has previously been shown to act as an attention sink [1], underscoring its importance in maintaining model accuracy.
>
> | **# Outliers**                | **N-S1**   | **N-S2**   | **N-MK1** | **N-MK2** | **N-MQ**  | **N-MV**  | **QA-1**  | **QA-2**  | **VT**    | **FWE**   | **Avg.**  |
> |---------------------------|--------|--------|-------|-------|-------|-------|-------|-------|-------|-------|-------|
> | 0 (0.000%)              | 100.00 | 100.00 | 96.88 | 85.42 | 73.18 | 70.83 | 43.75 | 39.58 | 73.54 | 57.29 | 74.05 |
> | 1 (0.006%)              | 100.00 | 100.00 | 97.92 | 98.96 | 95.83 | 94.79 | 70.83 | 51.04 | 70.63 | 70.14 | 85.01 |
> | 2 (0.012%)              | 100.00 | 100.00 | 97.92 | 98.96 | 95.57 | 95.57 | 70.83 | 51.04 | 72.08 | 70.49 | 85.25 |
> | 4 (0.024%)              | 100.00 | 100.00 | 97.92 | 98.96 | 95.83 | 95.57 | 71.88 | 51.04 | 74.38 | 71.18 | 85.68 |
> | 8 (0.049%)              | 100.00 | 100.00 | 97.92 | 98.96 | 95.57 | 95.05 | 72.92 | 51.04 | 78.13 | 72.57 | 86.22 |
> | 16 (0.098%)             | 100.00 | 100.00 | 97.92 | 98.96 | 96.09 | 95.31 | 72.92 | 51.04 | 80.42 | 71.53 | 86.42 |
> | 32 (0.195%)             | 100.00 | 100.00 | 97.92 | 98.96 | 96.35 | 95.57 | 72.92 | 52.08 | 81.25 | 72.22 | 86.73 |
> | 48 (0.293%)             | 100.00 | 100.00 | 97.92 | 98.96 | 96.88 | 95.83 | 72.92 | 52.08 | 81.67 | 72.57 | 86.88 |
> | *Quest (Ref.)*     | 100.00 | 100.00 | 98.96 | 77.08 | 97.65 | 93.49 | 60.42 | 50.00 | 77.08 | 65.63 | 82.03 |
> | *Full Attn (Ref.)* | 100.00 | 100.00 | 98.96 | 98.96 | 98.96 | 95.57 | 75.00 | 48.96 | 78.54 | 71.85 | 86.68 |
>
>
> The results demonstrate that increasing the number of outlier chunks has a positive impact on accuracy, especially in complex tasks. This indicates that even a small number of outliers can effectively capture essential information, reducing the need for full attention. **Remarkably, with just 8 outliers (0.049%), ShadowKV outperforms the Quest baseline and nearly matches the accuracy achieved by full attention.**
>
> However, when outliers are not adequately managed, the performance of the mean-based landmarks in ShadowKV may fall below the min-max approach used by Quest, underscoring the importance of handling outliers properly.
>
> [1] Efficient Streaming Language Models with Attention Sinks

---

> ### Author Response · Authors · 2024-11-20
> **Response to Reviewer 4R5o (Part 2/4)**
>
> **Q2: Detailed Efficiency Breakdown Across System Components**
>
> Thank you for the valuable suggestion. We present a detailed latency breakdown in tables below to illustrate the efficiency of each operation under various context lengths for both the prefilling and decoding stages (mentioned in $\textcolor{blue}{\text{Section 5.2, Page 9}}$ and detailed in  $\textcolor{blue}{\text{Appendix A.6, Page 18}}$).
>
> - **Scalability for Longer Sequences.** As shown in table, the overhead of SVD, reduce, cosine similarity, topK, and gather computing is very low and tends to decrease as the sequence scales, proving that ShadowKV's scalability to longer sequences.
>
> Latency breakdown (ms) of a Transformer block of Llama-3-8B-1M during prefilling:
>
> | **Context** | **Attention** | **FFN**    | **SVD**    | **Reduce** | **CosineSimilarity** | **TopK** | **Gather** | **Cost**   |
> |---------|-----------|--------|--------|--------|------------------|------|--------|--------|
> | 64K     | 186.23    | 96.47  | 17.19  | 0.10   | 1.41             | 0.08 | 0.01   | 6.65% |
> | 128K    | 721.13    | 193.32 | 26.62  | 0.20   | 2.77             | 0.14 | 0.02   | 3.25% |
> | 256K    | 2880.21   | 392.77 | 50.56  | 0.42   | 6.11             | 0.11 | 0.03   | 1.75% |
> | 512K    | 11720.30  | 789.23 | 108.38 | 0.84   | 12.19            | 0.15 | 0.06   | 0.97% |
>
> - **Overlapping Operations for Latency Reduction.** In the table below, we demonstrate how overlapping the recomputation of the key cache with value cache fetching from the CPU significantly reduces decoding latency. This concurrent processing approach ensures that ShadowKV minimizes overhead when handling long-context models.
>
> Latency breakdown (ms) of a Transformer block of Llama-3-8B-1M during decoding:
>
> | **Context**        | **GEMM+Softmax** | **Max** | **TopK**| **Recompute K (Overlapped)** | **Fetch V** | **Attention** | **FFN** | **QKV**|
> |----------------|---------------------------------|-----------|------------|--------------------------------------|---------------|-----------------|-----------|-----------|
> | 48$\times$64K   | 0.56                            | 0.07      | 0.14       | 1.25                                 | 1.84          | 0.23            | 0.33      | 0.05      |
> | 24$\times$128K | 0.58                            | 0.07      | 0.15       | 1.36                                 | 1.66          | 0.21            | 0.29      | 0.05      |
> | 12$\times$256K | 0.65                            | 0.07      | 0.16       | 1.49                                 | 1.75          | 0.19            | 0.25      | 0.05      |
> | 6$\times$512K  | 0.71                            | 0.07      | 0.17       | 1.51                                 | 1.69          | 0.18            | 0.23      | 0.05      |

---

> ### Author Response · Authors · 2024-11-20
> **Response to Reviewer 4R5o (Part 3/4)**
>
> **Q3: Performance on Extremely Long Context Lengths**
>
> We clarify that our paper includes some experiments with extremely long contexts. We tested Llama-3-8B-1M on the NIAH dataset with up to 1M tokens ($\textcolor{blue}{\text{Figure 6, Page 8}}$).
>
> Here we present our additional results, showing that ShadowKV matches the performance of full attention while outperforming other sparse methods on 1M contexts with Llama-3-8B-1M and 512K contexts with Llama-3-70B-1M, as demonstrated on the RULER benchmark (detailed in $\textcolor{blue}{\text{Section 5.1, Page 7}}$ and $\textcolor{blue}{\text{Appendix A.5, Page 17}}$). Additionally, ShadowKV achieves perfect needle retrieval accuracy with Llama-3-70B-1M, evaluated across context lengths ranging from 16K to 1M tokens ($\textcolor{blue}{\text{Figure 10, Page 17}}$).
>
> |**Methods**|**N-S1**|**N-S2**|**N-MK1**|**N-MK2**|**N-MQ**|**N-MV**|**QA-1**|**QA-2**|**VT**|**FWE**|**Avg.**|
> |-------------------------|-----------------|-----------------|----------------|----------------|----------------|----------------|----------------|----------------|----------------|----------------|----------------|
> |*Llama-3-70B-1M*|100.00|82.29|90.63|54.17|85.16|96.61|69.79|35.42|68.75|69.44|75.23|
> |Loki|100.00|1.04|0.00|0.00|0.00|0.00|13.54|11.46|34.30|22.92|18.33|
> |Loki (Vonly)|100.00|15.63|26.04|0.00|0.00|0.00|25.00|19.79|40.00|31.94|25.84|
> |Quest|100.00|76.04|78.13|35.42|85.47|92.19|53.21|34.38|38.33|58.33|65.15|
> |Quest (Vonly)|100.00|77.08|79.17|36.49|86.19|**95.31**|54.17|36.58|47.70|58.68|67.14|
> |**ShadowKV**|**100.00**|**82.29**|**88.54**|**53.04**|**88.02**|94.79|**67.71**|**37.50**|**68.54**|**68.25**|**74.87**|
> |*Llama-3-8B-1M*|96.88|100.00|96.88|69.79|91.15|85.68|64.58|42.71|25.00|56.25|72.89|
> |Loki|9.38|1.04|10.42|0.00|2.60|4.43|38.54|11.46|1.67|0.69|8.02|
> |Loki (Vonly)|68.75|29.17|60.42|1.04|26.56|43.23|59.38|15.63|6.46|0.69|31.13|
> |Quest|94.79|92.71|80.21|4.17|76.30|69.27|57.29|28.13|25.67|30.56|55.91|
> |Quest (Vonly)|94.79|93.75|81.25|4.17|79.69|69.27|62.50|31.25|26.00|32.99|57.57|
> |**ShadowKV**|**96.88**|**100.00**|**96.88**|**65.63**|**89.38**|**83.16**|**69.79**|**42.71**|**26.04**|**59.38**|**72.98**|
>
> These findings underline ShadowKV’s ability to handle large-scale inputs effectively, offering robust performance across increasing context lengths and model sizes. This scalability ensures its suitability for real-world applications that require extensive sequence handling and larger model capabilities.

---

> > ### Author Response · Authors · 2024-11-20
> > **Response to Reviewer 4R5o (Part 4/4)**
> >
> > **Q4: Efficiency Comparison with Prior Works**
> >
> > Thank you for the suggestion. We first clarify why we use sparse budgets in the original draft to compare the efficiency and accuracy of ShadowKV against the baselines, followed by an explanation of MInference’s orthogonal relationship to our method.
> >
> > - Since Loki and Quest lack CPU offloading and optimized CUDA kernels in their open-source repositories, we ensured a fair comparison by standardizing the computational cost for KV selection and using the same sparse KV cache budget across methods. This budget represents the theoretical efficiency of each method, allowing for a fair comparison of accuracy on downstream tasks.
> > - For MInference, we clarify that it is intended to accelerate prefilling rather than decoding, and we have demonstrated its compatibility with our approach in Table 3 in the paper.
> >
> > In order to compare the efficiency of ShadowKV against Quest, we try our best to implement Quest with CPU offloading and present the result ($\textcolor{blue}{\text{Appendix A.7, Page 18}}$), showing that **ShadowKV achieves up to 4.85$\times$ faster performance than Quest**.
> >
> > | **Context**     | **Full Attention** | **Full Attention (CPU-offload)** | **Quest** | **Quest (CPU-offload)**   | **ShadowKV**           |
> > |-------------|----------------|----------------------|-------|---------------|----------------|
> > | 3$\times$1M | OOM            | 0.21 tokens/s        | OOM   | 9.34 tokens/s | **45.32 tokens/s** |
> >
> > As shown in the tale, ShadowKV significantly outperforms both Full Attention and Quest under the same sparse budget. The efficiency advantage of ShadowKV over Quest is due to two key factors:
> > - ShadowKV only fetches the value cache from the CPU, rather than the entire KV pair, minimizing data transfer and reducing latency
> >  - ShadowKV integrates a cache mechanism that leverages the temporal locality of the KV cache.

---

> > > ### Author Response · Authors · 2024-11-25
> > > **Grateful for Your Insights and Follow-Up on Our Responses for Reviewer 4R5o**
> > >
> > > Dear Reviewer 4R5o,
> > >
> > > We sincerely appreciate the time and effort you have dedicated to reviewing our work and for sharing your insightful feedback. Your comments have been immensely valuable in improving the quality of our manuscript.
> > >
> > > We have carefully addressed your concerns and revised the manuscript accordingly. As the discussion period is ending soon, we wanted to kindly ask if there are any further questions or points we could clarify to better address your feedback. If our responses have adequately addressed your concerns, we would be deeply grateful if you could consider reflecting this in your score.
> > >
> > > Thank you again for your constructive review and for contributing your valuable expertise. We deeply appreciate your input and support throughout this process.

---

> > > > ### Comment · Reviewer_4R5o · 2024-11-25
> > > >
> > > > Thank you for your extensive experiments, which have addressed most of my concerns. I have increased my score accordingly.

---

### Official Review · Reviewer_sEj1 · 2024-11-04

**Soundness:** 3
**Presentation:** 3
**Contribution:** 3
**Rating:** 8
**Confidence:** 4

**Summary:**

This paper presents a high throughput LLM inference system for long sentence length. It proposes a compression method to leverage the low rank property of key cache and improve the sparse attention method by accurate KV selection.

**Strengths:**

This paper is well-organized and clearly written. The proposed method is well-motivated, addressing relevant challenges, and is supported by thorough analysis. The evaluation is comprehensive and robust, effectively substantiating the claims and demonstrating thoughtful considerations. Overall, this is a strong submission for ICLR.

**Weaknesses:**

1. As shown in Fig. 8, some downstream tasks, such as 'Frequent Words Extraction,' perform significantly worse with sparse KV enabled. A brief analysis of why this approach underperforms for these types of tasks would be helpful, as well as any potential solutions to address these limitations.
2. The proposed solution is currently evaluated on an 8B model with a 128K sequence length. It would strengthen the paper to include an analysis of whether this approach scales effectively for larger models, such as a 70B model with an extremely long sequence length of 1M.

**Questions:**

As listed in the weakness part.

---

> ### Author Response · Authors · 2024-11-20
> **Response to Reviewer sEj1 (Part 1/2)**
>
> Thank you for the supportive comments and recognizing the novelty of our method and the thorough evaluations. We hope our detailed clarifications and additional experimental results will address the concerns regarding our work.
>
> **Q1: Analysis and Solutions for Sparse KV Underperformance in Certain Tasks**
>
> For certain downstream tasks, attention distributions can exhibit a long tail effect, where a larger portion of tokens receive small yet non-negligible attention, making exact top-K selection less effective. As observed in Differential Transformer [1], this issue arises due to the nature of the softmax function [2], which tends to produce noise and disperse attention scores.
>
> Since Top-K serves as the theoretical upper bound for sparse attention methods aiming to approximate it, poor performance of Top-K inherently limits the performance of sparse attention methods.
>
> We include TopK results for ‘Frequent Words Extraction’ as a reference below. As shown in the table, sparse attention underperforms due to limitations in TopK’s performance.
>
> | **Method**       | **Full (128K)** | **4096**  | **2048**  | **1024**  | **512**   | **256**   | **128**   |
> | ------------ | ----------- | ----- | ----- | ----- | ----- | ----- | ----- |
> | TopK         | 72.22       | 69.21 | 68.37 | 67.32 | 67.01 | 66.59 | 62.85 |
> | **ShadowKV** | 72.22       | 68.06 | 68.06 | 68.40 | 66.32 | 65.28 | 59.72 |
> | Quest        | 72.22       | 66.67 | 65.97 | 64.58 | 62.50 | 61.81 | 50.69 |
>
>
> A possible solution is to promote sparsity during the pre-training or SFT phases, co-designing it with the training process. By encouraging sparsity throughout training, the model may learn to manage long tail effects within the constraints of a sparse budget and reduce noise, allowing for a more reliable sparse attention mechanism across diverse downstream tasks.
>
> [1] Differential Transformer
>
> [2] softmax is not enough (for sharp out-of-distribution)

---

> > ### Author Response · Authors · 2024-11-20
> > **Response to Reviewer sEj1 (Part 2/2)**
> >
> > **Q2: Scalability Analysis for Larger Models and Longer Sequence Lengths**
> >
> > We appreciate the suggestion to evaluate ShadowKV on larger models with extended sequence lengths, such as 1M. Our results show that ShadowKV matches the performance of full attention while outperforming other sparse methods on 1M contexts with Llama-3-8B-1M and 512K contexts with Llama-3-70B-1M, as demonstrated on the RULER benchmark (mentioned in $\textcolor{blue}{\text{Section 5.1, Page 7}}$ and detailed in $\textcolor{blue}{\text{Appendix A.5, Page 17}}$). Additionally, ShadowKV achieves perfect needle retrieval accuracy with Llama-3-70B-1M, evaluated across context lengths ranging from 16K to 1M tokens ($\textcolor{blue}{\text{Figure 10, Page 17}}$).
> >
> >
> > |**Methods**|**N-S1**|**N-S2**|**N-MK1**|**N-MK2**|**N-MQ**|**N-MV**|**QA-1**|**QA-2**|**VT**|**FWE**|**Avg.**|
> > |-------------------------|-----------------|-----------------|----------------|----------------|----------------|----------------|----------------|----------------|----------------|----------------|----------------|
> > |*Llama-3-70B-1M*|100.00|82.29|90.63|54.17|85.16|96.61|69.79|35.42|68.75|69.44|75.23|
> > |Loki|100.00|1.04|0.00|0.00|0.00|0.00|13.54|11.46|34.30|22.92|18.33|
> > |Loki (Vonly)|100.00|15.63|26.04|0.00|0.00|0.00|25.00|19.79|40.00|31.94|25.84|
> > |Quest|100.00|76.04|78.13|35.42|85.47|92.19|53.21|34.38|38.33|58.33|65.15|
> > |Quest (Vonly)|100.00|77.08|79.17|36.49|86.19|**95.31**|54.17|36.58|47.70|58.68|67.14|
> > |**ShadowKV**|**100.00**|**82.29**|**88.54**|**53.04**|**88.02**|94.79|**67.71**|**37.50**|**68.54**|**68.25**|**74.87**|
> > |*Llama-3-8B-1M*|96.88|100.00|96.88|69.79|91.15|85.68|64.58|42.71|25.00|56.25|72.89|
> > |Loki|9.38|1.04|10.42|0.00|2.60|4.43|38.54|11.46|1.67|0.69|8.02|
> > |Loki (Vonly)|68.75|29.17|60.42|1.04|26.56|43.23|59.38|15.63|6.46|0.69|31.13|
> > |Quest|94.79|92.71|80.21|4.17|76.30|69.27|57.29|28.13|25.67|30.56|55.91|
> > |Quest (Vonly)|94.79|93.75|81.25|4.17|79.69|69.27|62.50|31.25|26.00|32.99|57.57|
> > |**ShadowKV**|**96.88**|**100.00**|**96.88**|**65.63**|**89.38**|**83.16**|**69.79**|**42.71**|**26.04**|**59.38**|**72.98**|
> >
> > These findings underline ShadowKV’s ability to handle large-scale inputs effectively, offering robust performance across increasing context lengths and model sizes. This scalability ensures its suitability for real-world applications that require extensive sequence handling and larger model capabilities.

---

> > > ### Comment · Reviewer_sEj1 · 2024-11-21
> > >
> > > Thank you for your responses. The rebuttal has clearly address my concerns on the Underperformance in Certain Tasks and scalability problem.

---

### Author Response · Authors · 2024-11-20
**Revised Manuscript updated by Authors**

We thank reviewers [R1(sEj1), R2(4R5o), R3(5wSX), R4(iRxd)] for their thoughtful and highly supportive feedback! We were glad that the reviewers found the problem **significant and interesting [R2, R4]**, the observations and analysis **insightful and highly valuable [R1, R2, R3]**, the method **novel and well-motivated [R1, R3, R4]**, the presentation **well-organized and easy to follow [R1]**, experimental results **strong and impressive [R1, R3, R4]**.

We have updated the paper to incorporate the constructive suggestions, reflected in the revised version. We use **blue color for all newly added content**. Here is a summary of the major changes:

- [R1, R2, R3] **Scalability to larger models, longer contexts, and longer outputs**:
	- Added experiments for Llama-3-70B-1M on RULER with 512K contexts and Needle In A Haystack with up to 1M contexts, demonstrating ShadowKV’s scalability with larger models (mentioned in $\textcolor{blue}{\text{Section 5.1, Page 7}}$ and detailed in $\textcolor{blue}{\text{Appendix A.5, Page 17}}$).
	- Added experiments for Llama-3-8B-1M on RULER with 1M contexts to show ShadowKV’s ability to scale with longer sequences (mentioned in $\textcolor{blue}{\text{Section 5.1, Page 7}}$ and detailed in $\textcolor{blue}{\text{Appendix A.5, Page 17}}$).
	- Added experiments for handling newly generated tokens to validate ShadowKV’s scalability for longer outputs (mentioned in $\textcolor{blue}{\text{Section 4.1, Page 6}}$ and detailed in  $\textcolor{blue}{\text{Appendix A.1, Page 15}}$).
- [R3] **Comparison with InfiniGen**
	- Added a discussion of the differences between InfiniGen and ShadowKV ($\textcolor{blue}{\text{Section 2, Page 3}}$).
	- Included comparative experiments with InfiniGen, showing that ShadowKV outperforms InfiniGen on various downstream tasks. Notably, on the RULER benchmark, ShadowKV achieves an accuracy improvement of 10% over InfiniGen ($\textcolor{blue}{\text{Section 5.1, Page 7-8}}$).
- [R2, R4] **Latency breakdown**
	- Added a latency breakdown for the prefilling stage, demonstrating that ShadowKV introduces minimal overhead, which decreases as sequence length increases, affirming ShadowKV’s scalability with longer sequences (mentioned in $\textcolor{blue}{\text{Section 5.2, Page 9}}$ and detailed in  $\textcolor{blue}{\text{Appendix A.6, Page 18}}$).
	- Added a latency breakdown for the decoding stage, showing how overlapping the recomputation of the key cache with value cache fetching from the CPU effectively reduces decoding latency (mentioned in $\textcolor{blue}{\text{Section 5.2, Page 9}}$ and detailed in  $\textcolor{blue}{\text{Appendix A.6, Page 18}}$).
- [R4] **Numerical stability**
	- Added experiments to show that ShadowKV can retain its accuracy at lower precision, e.g., FP8 (mentioned in $\textcolor{blue}{\text{Section 5.3, Page 10}}$ and detailed in  $\textcolor{blue}{\text{Section A.4, Page 16}}$).
- [R2] **Extra ablations**
	- Added experiments demonstrating the effectiveness of outliers and mean landmarks (compared to Quest’s min-max landmarks), highlighting the improved performance of our method (mentioned in $\textcolor{blue}{\text{Section 5.3, Page 10}}$ and detailed in  $\textcolor{blue}{\text{Appendix A.8, Page 19}}$).
	- Included an efficiency comparison with Quest, showing that our method outperforms Quest in both accuracy and efficiency. Specifically, ShadowKV is up to 4.85$\times$ faster than Quest while also achieving superior accuracy ($\textcolor{blue}{\text{Appendix A.7, Page 18}}$).

---

### Meta-Review · Area_Chair_RRQd · 2024-12-15

**Metareview:**

After reviewing the entire discussion and considering input from all reviewers, I recommend the rejection of this paper. While the submission has merits, the unresolved concerns outweigh the strengths.

**Strengths**:
- The paper presents a novel observation about efficiently compressing pre-RoPE Key caches, reducing CPU-GPU communication costs. This is a potentially useful insight that has practical implications for offloading methods.
- The revised methodology demonstrates improved scalability for long sequences while maintaining accuracy.

**Weaknesses**:
1. **Incremental Contribution**:
   The proposed work builds on existing methods of CPU offloading and low-rank approximation. However, the enhancements, while effective, appear to be incremental rather than groundbreaking.

2. **Presentation and Positioning**:
   The paper struggles with positioning itself clearly within the existing literature. The unique contributions remain insufficiently distinguished from prior work, leaving ambiguity about its originality. This issue undermines the perceived value of the paper's contributions.


Given these mixed reviews, the lack of consensus among reviewers, and the remaining issues of novelty and presentation, I believe the paper does not meet the threshold for acceptance in its current form. While the work demonstrates promise, a more substantial contribution and clearer articulation of its impact are necessary to justify publication.

**Additional Comments On Reviewer Discussion:**

- Reviewer sEj1 supported the paper, citing its strong experimental results and the practical implications of its key cache compression technique. However, even this reviewer acknowledged the need for better articulation of the contributions.
- Reviewer iRxd expressed moderate support but did not strongly champion the paper and provided limited commentary on its novelty or positioning.
- Reviewer 5wSX maintained a neutral stance, citing the incremental nature of the contributions and unclear positioning as significant concerns.

---

### Decision · Program_Chairs · 2025-01-22

Reject